# Reactive Oxygen Species-Mediated Autophagy by Ursolic Acid Inhibits Growth and Metastasis of Esophageal Cancer Cells

**DOI:** 10.3390/ijms21249409

**Published:** 2020-12-10

**Authors:** Na-Ri Lee, Ruo Yu Meng, So-Young Rah, Hua Jin, Navin Ray, Seong-Hun Kim, Byung Hyun Park, Soo Mi Kim

**Affiliations:** 1Division of Hematology and Oncology, Jeonbuk National University Medical School, Jeonju 54907, Korea; nariflower@jbnu.ac.kr; 2Department of Internal Medicine, Jeonbuk National University Medical School, Jeonju 54907, Korea; shkimgi@jbnu.ac.kr; 3Research Institute of Clinical Medicine, Biomedical Research Institute of Jeonbuk National University Medical School, Jeonju 54907, Korea; 4Department of Physiology and Institute of Medical Science, Jeonbuk National University Medical School, Jeonju 54907, Korea; kathymeng1216@gmail.com (R.Y.M.); navin.ray@gmail.com (N.R.); 5Department of Biochemistry, Jeonbuk National University Medical School, Jeonju 54907, Korea; syrah1004@hanmail.net (S.-Y.R.); bhpark@jbnu.ac.kr (B.H.P.); 6School of Pharmaceutical Sciences, Tsinghua University, Beijing 100084, China; jinhuaxy@126.com

**Keywords:** ursolic acid, esophageal squamous cell carcinoma, anticancer, autophagy, reactive oxygen species, cell death

## Abstract

Ursolic acid (UA) possesses various pharmacological activities, such as antitumorigenic and anti-inflammatory effects. In the present study, we investigated the mechanisms underlying the effects of UA against esophageal squamous cell carcinoma (ESCC) (TE-8 cells and TE-12 cells). The cell viability assay showed that UA decreased the viability of ESCC in a dose-dependent manner. In the soft agar colony formation assay, the colony numbers and size were reduced in a dose-dependent manner after UA treatment. UA caused the accumulation of vacuoles and LC3 puncta, a marker of autophagosome, in a dose-dependent manner. Autophagy induction was confirmed by measuring the expression levels of LC3 and p62 protein in ESCC cells. UA increased LC3-II protein levels and decreased p62 levels in ESCC cells. When autophagy was hampered using 3-methyladenine (3-MA), the effect of UA on cell viability was reversed. UA also significantly inhibited protein kinase B (Akt) activation and increased p-Akt expression in a dose-dependent manner in ESCC cells. Accumulated LC3 puncta by UA was reversed after wortmannin treatment. LC3-II protein levels were also decreased after treatment with Akt inhibitor and wortmannin. Moreover, UA treatment increased cellular reactive oxygen species (ROS) levels in ESCC in a time- and dose-dependent manner. Diphenyleneiodonium (an ROS production inhibitor) blocked the ROS and UA induced accumulation of LC3-II levels in ESCC cells, suggesting that UA-induced cell death and autophagy are mediated by ROS. Therefore, our data indicate that UA inhibits the growth of ESCC cells by inducing ROS-dependent autophagy.

## 1. Introduction

Esophageal cancer is the eighth most prevalent cancer worldwide and the sixth main cause of cancer-related death, resulting in 456,000 new cases per year [1,2] Esophageal squamous cell carcinoma (ESCC) is predominant in Asian countries, and poor prognoses are often observed after ESCC resection because of its location [3,4,5,6]. The proven risk factors for ESCC are alcohol consumption and tobacco use [7]; however, acknowledged risk factors include low consumption of fruits and vegetables, zinc or vitamin E deficiency, and poor oral hygiene, which can be attributed to low socioeconomic status and living in a developing country [8,9]. ESCC is a heterogeneous disease with variable outcomes that are challenging to predict [10,11]. The best curative method for ESCC is surgery; however, complete surgical removal of ESCC is difficult in patients with late-stage ESCC [12,13], resulting in a high incidence of tumor recurrence. In addition, the absence of a molecular classification system and the clinical heterogeneity of ESCC have hindered the development of treatment standards. Therefore, there is a pressing need for novel therapeutic approaches to treat ESCC.

Natural compounds have been widely employed as therapeutic agents against cancer. Triterpenoids are universally found in plants; specifically, pentacyclic triterpenes have distinctive biological properties that have been demonstrated in recent studies [14]. Ursolic acid (UA), a pentacyclic triterpenoid compound found in several fruits and plants, has a distinctive therapeutic property [15]. Plants with significant UA content include apples (specifically the peel) and the leaves of oregano, rosemary, sage, eucalyptus, lavender, and coffee [16]. Several studies have shown that UA possesses anti-proliferative, pro-apoptotic, anti-inflammatory, anti-metastatic, and antiangiogenic properties against cancer cells [17,18,19,20,21,22,23]. UA inhibits cell proliferation, arrests cell cycle progression, and reduces tumorigenesis through multiple signaling pathways [24,25,26,27,28,29,30,31,32,33]. Pentacyclic triterpenoid compounds have been found to display their anticancer properties by altering the levels of reactive oxygen species (ROS), derivatives of oxygen metabolism that are generated under conditions of cellular oxidative stress. ROS play an important role in maintaining homeostasis and can have both advantageous and disadvantageous effects on cells. Esophageal cancer cells generate moderate ROS levels to aid in their proliferation, migration, and metastasis [34,35,36,37]. UA suppresses oxidative stress via liver kinase B1-adenosine monophosphate-activated protein kinase signaling [38]. However, the biological function of UA that affects ROS levels in ESCC has not been explored to date.

Autophagy is an evolutionarily conserved process responsible for cellular physiology and homeostasis; it is activated under physiological or chemical stress and degrades unwanted and harmful proteins through lysosomal degradation of autophagosomes [39], a phenomenon involved in tumorigenesis. Autophagy can kill the cell; therefore, activation of the autophagic cell death program is evolving as a potential cancer therapy. UA has been found to induce autophagy in various cancer cells, causing autophagic cell death [21,40]. Leng et al. showed that autophagy induction by UA killed cervical cancer (TC1) cells [40]. Moreover, UA mediated its anticancer effect via targeting several signaling pathways in breast cancer cells [21,41]. Xavier et al. also reported that UA induces cell death and modulates autophagy through the c-Jun N-terminal kinase pathway in apoptosis-resistant colorectal cancer cells [42]. The amassing of ROS leads to autophagy, and p62 plays an important role in the degradation of oxidized proteins in autophagosomes [43]. However, the molecular mechanisms underlying the effects of UA in ROS-mediated autophagy in ESCC have not been clearly elucidated yet. Considering that UA-induced autophagy has been implicated in various human cancers, an exploration of the novel signaling pathways relevant to ROS-mediated autophagy would shed light on the development of new therapeutic strategies for ESCC. To address these issues, we investigated whether UA affected ROS levels through autophagy in human ESCC cells. We demonstrated that UA induced autophagy in esophageal cancer cells by elevating ROS levels via the protein kinase B (Akt) signaling pathway. Therefore, this study showed that UA exerts an antitumorigenic effect on ESCC cells.

## 2. Results

### 2.1. UA Inhibited the Proliferation of ESCC Cells

To evaluate the effect of UA on the viability of ESCC cells, we performed an MTT cell viability assay. TE-8 and TE-12 cells were treated with various concentrations of UA (0–50 µM) for 48 h. The IC50 of TE-8 was 39.01 µM, and the IC50 of TE-12 was 29.65 µM. The MTT assay results showed that UA significantly decreased the ESCC viability in a dose-dependent manner (Figure 1A). The cell viability data show that UA induced more than 50% cell death at a concentration of 30 µM in TE-12 cells and 40 µM in TE-8 cells. In soft agar colony formation assay, the number and size of colonies were reduced in a dose-dependent manner after UA treatment, indicating the anti-proliferative property of UA against growth of TE-8 and TE-12 cells (Figure 1B). To examine whether UA causes apoptosis in esophageal cancer cells, we measured the protein levels of cleaved-caspase-3, caspase-3, cleaved-poly [ADP-ribose] polymerase (PARP), and PARP in TE-8 and TE-12 cells. As shown in Figure 1C, UA suppressed PARP and caspase-3 protein levels and increased the protein levels of the cleaved form of PARP and cleaved-caspase-3 in TE-8 and TE-12 cells. To further confirm the induction of apoptosis by UA in esophageal cancer cells, we performed a flow cytometry analysis. Annexin V staining results showed that both the early stage of apoptotic cells and the late stage of apoptotic cells were increased after UA treatment in a dose-dependent manner in the TE-8 and TE-12 cells (Figure 1D). Moreover, sub-G1 phase cells are considered as apoptosis cells, whose number we calculated using a cell cycle analysis. UA significantly induced sub-G1 phase cells in a dose-dependent manner in esophageal cancer cells (TE-8 and TE-12 cells) (Figure 1E). These results indicate that UA induced caspase-dependent cell death in ESCC cells.

### 2.2. UA Suppressed the Migration and Invasion of ESCC Cells

Migration and invasion are the initial and critical events in metastasis [44]. To evaluate the migratory capacity of TE-8 and TE-12 cells under the influence of UA, we conducted a wound healing assay. The scratch was made using a 200 µL pipette tip on a monolayer culture of the TE-8 and TE-12 cells. The migratory capacity of the TE-8 and TE-12 cells treated with UA was remarkably decreased in a dose-dependent manner. The wound treated with 50 µM of UA healed more slowly than the wound treated with 0, 10, and 25 µM of UA (Figure 2A,B). We further investigated the invasion ability of the TE-8 and TE-12 cells after UA treatment. As shown in Figure 2C, the Matrigel invasion assay demonstrated significantly decreased invasion effects after UA in the TE-8 and TE-12 cells in a dose-dependent manner. The results suggest that UA inhibits the migration and invasion of ESCC cells.

### 2.3. UA Caused Induction of Autophagy in ESCC Cells

Studies have reported that UA induces autophagy in a variety of cancer cells such as cervical cancer cells and breast cancer cells [21,40]. Therefore, we investigated whether UA induces autophagy in ESCC cells. UA accumulated vacuoles in cytoplasm and green fluorescent protein (GFP)-tagged LC3 puncta, a marker of autophagosome, in a dose-dependent manner (Figure 3A). The visualization of the transfected GFP-tagged LC3 puncta in the TE-12 cells under a confocal microscope appeared stronger in the cells treated with higher UA concentrations than in the GFP-LC3 transfected cells (Figure 3A). Autophagy induction was confirmed by measuring the expression levels of LC3 and p62 protein in the ESCC cells. As shown in Figure 3B, UA increased LC3-II protein levels and decreased p62 protein levels in a dose-dependent manner in the TE-8 and TE-12 cells. These data suggested that UA induced autophagy in the ESCC cells in a dose-dependent manner. Given that 3-methyladenine (3-MA) is an autophagy inhibitor, we employed 3-MA to further investigate the role of UA in autophagy in ESCC cell proliferation. Autophagy inhibition through pretreatment with 3-MA (5 mM) and UA treatment (30 µM) resulted in increased ESCC cell survival. UA-induced inhibition of cell viability was recovered by 3-MA (5 mM) in the TE-8 and TE-12 cells (Figure 3C). In addition, autophagy inhibition by 3-MA caused LC3 protein expression and p62 alteration. The TE-8 and TE-12 cells treated with a combination of UA (30 µM) and 3-MA (5 mM) promoted the expression of p62 and LC3-II compared to those treated with UA (30 µM) alone (Figure 3D). These data therefore indicate that UA might be responsible for ESCC cell death by inducing autophagy.

### 2.4. Ursolic Acid Caused Induction of Akt-Dependent Autophagy in ESCC Cells

The target of rapamycin (TOR), a downstream component of the phosphatidylinositol 3-kinase (PI3K)/Akt pathway, is critical for autophagy. The upstream signal PI3K/Akt activates TOR, thereby suppressing autophagy [45]. To further determine whether UA-induced autophagy is mediated through the Akt-mTOR signaling pathway, we analyzed the protein levels of the Akt-mTOR signaling components. The results of the Western blotting showed that UA inhibited the phosphorylation of Akt protein in a dose-dependent manner (Figure 4A). The protein levels of mTOR and Bcl-2 were reduced, whereas those of Bax and Beclin-1 were increased after 48 h of UA treatment in a dose-dependent manner in the TE-8 and TE-12 cells (Figure 4B). However, class III PI3K is involved in the formation of autophagosomes, and treatment with wortmannin, a PI3K inhibitor, inhibits autophagy [46]. Our data show that the accumulated vacuoles in the cytoplasm and the GFP-tagged LC3 puncta through UA treatment were inhibited after wortmannin treatment (Figure 4C). Furthermore, autophagy inhibition by wortmannin or the Akt inhibitor was confirmed by measuring the expression levels of LC3 and p62 proteins in the ESCC cells. As shown in Figure 4D, the increased LC3-II expression by UA treatment was inhibited after treatment with Akt inhibitor or wortmannin along with UA in the TE-8 and TE-12 cells. These data therefore suggest that UA-induced autophagy resulted in inhibited cell survival mediated through the Akt-mTOR signaling pathway.

### 2.5. Ursolic-Acid-Induced Autophagy Mediation by ROS Production

ROS are related to the formation of autophagosomes, and UA induces ROS production in human glioma cells [47]. To further investigate whether autophagy induction by UA is due to ROS production in ESCC cells, we measured the dichlorofluorescein diacetate (DCFDA) fluorescence after UA treatment. The confocal fluorescence microscopy image showed that the TE-8 and TE-12 cells treated with UA (30 µM) resulted in increased intracellular DCFDA fluorescence in a time-dependent manner (Figure 5A). These data indicate that ROS production was increased by UA treatment in a time-dependent manner in the ESCC cells. As shown in Figure 5B, the DCFDA fluorescence was increased after UA treatment in a dose-dependent manner in the TE-8 and TE-12 cells. The induced DCFDA fluorescence after UA treatment was inhibited in the presence of 100 nM of diphenyleneiodonium (DPI) (a ROS production inhibitor) and 1 mM of *N*-acetyl-cysteine (NAC; an ROS production inhibitor) in the TE-8 and TE-12 cells (Figure 5C). We further assessed cell proliferation and LC3 protein levels after DPI and UA treatment in the TE-8 and TE-12 cells. As shown in Figure 5D, the decreased cell viability after UA (30 µM) treatment was recovered by the DPI treatment in a dose-dependent manner in both the TE-8 and TE-12 cells. In addition, the increased LC3-II protein levels by UA were also inhibited after DPI treatment in a dose-dependent manner in the TE-8 cells; however, these protein levels were only inhibited by a high dose of DPI (200) µM in the TE-12 cells (Figure 5E). Taken together, these data indicate that UA might be responsible for the autophagy of ESCC by increasing ROS levels.

## 3. Discussion

Esophageal cancer has a high mortality rate because of its biological aggressiveness [2]. Considering the cancer’s characteristics, the patient survival rate for esophageal cancer is low [4,5,6]. The incidence of esophageal cancer is growing because of increased consumption of Western diets, increasing obesity rate, and aging of the population in recent years [8]. Therefore, the development of new therapeutic drugs is essential for increasing patient survival. The present study was designed to investigate the mechanisms underlying the effects of UA in ESCC and to determine whether UA inhibits cell proliferation by inducing autophagy in esophageal cancer cells. We found that UA induced autophagy in ESCC by regulating ROS levels via the Akt-mTOR signaling pathway.

Natural compounds have been shown to have beneficial effects on a variety of cancer types [14]. UA, a pentacyclic triterpenoid compound found in herbs, spices, and the peels of fruits, has been shown to have anti-proliferative, pro-apoptotic, anti-inflammatory, anti-metastatic, and antiangiogenic properties against cancer cells [17,18,19,20,21,22,23]. Our study showed that UA significantly inhibited esophageal cancer cell growth in a dose-dependent manner and significantly attenuated colony formation. This apoptotic effect is consistent with the efficacy of UA in other cancers [26,27,28,29,30,31,32,33,34]. Therefore, UA has the effect of inhibiting cell growth in esophageal cancer. In addition, UA treatment markedly induced apoptosis by enhancing the levels of cleaved-PARP and caspase-9 proteins. The percentage of sub-G1 phase cells in both the early and late stages of cell apoptosis were increased after UA treatment, which also reduced cell migration and invasion in esophageal cancer cells in a dose-dependent manner. These observations are similar to the findings of previous studies, demonstrating that UA induced apoptosis and inhibited cell invasion and migration in various cancer cells [19,21,30,36,42,48]. Therefore, our present study provides strong evidence that UA inhibited cell growth and induced cell apoptosis in esophageal cancer cells.

Increasing evidence has shown that autophagy occurs through a lysosomal-mediated cellular self-digestion pathway and is directly involved in determining specific cell death [49]. The interaction between autophagy and cell death is an important mechanism for apoptosis- and necrosis-regulated cell death [50,51,52,53,54,55]. Our results showed that UA induced autophagy in a dose-dependent manner and markedly increased LC3-II levels and decreased p62 levels in esophageal cancer cells. The LC3-II protein levels increased by UA were suppressed by the autophagy inhibitor 3-MA, and the decreased p62 levels by UA were induced by 3-MA treatment. Moreover, the cell viability reduced by UA was recovered by 3-MA. These data provide strong evidence that UA induced autophagy in esophageal cancer cells. Several signaling transduction pathways of cancer have been implicated in autophagy, such as the Akt/mTOR pathway [56] and the Beclin 1/Bax pathway [57], and recent studies have reported that ROS are important triggers of autophagy under various circumstances [58,59]. Akt signaling is considered an oncogene because of its promoting effect on cell survival and inhibitory effect on apoptosis [60,61,62]. By stimulating oxidative metabolism, Akt also promotes mitochondrial oxygen consumption and contributes to ROS accumulation [63,64]. These studies suggest that the Akt-mTOR pathway is associated with the activation of autophagy, a finding that is consistent with our results showing that UA-induced autophagy was attenuated by Akt inhibitor treatment or wortmannin. In our study, UA treatment caused the inactivation of Akt-mTOR signaling, with Beclin 1 disconnecting from Bcl-2. UA increased LC3-II protein levels but inhibited the phosphorylation of Akt protein in a dose-dependent manner. After treatment with the Akt inhibitor or wortmannin and UA, the increased LC3-II expression by UA treatment was decreased. We have demonstrated that UA induced autophagy and autophagy-related cell death by regulating ROS levels via the Akt signaling pathway. However, there is still the question regarding the other important pathway: how does the Beclin 1/Bcl-2 complex regulate autophagy in ESCC after UA treatment? Further studies are needed to determine the involvement of the Beclin 1/Bcl-2 complex in UA-induced autophagy in ESCC cells.

Various anticancer agents have been demonstrated to effectively induce cell death by regulating ROS generation [65]. Shen et al. demonstrated that UA induces autophagy through ROS-mediated endoplasmic reticulum stress [47]. Consistent with these studies, we demonstrated that ROS levels were increased by UA treatment in a dose- and time-dependent manner in TE-8 and TE-12 cells. In addition, ROS production and LC3-II protein levels induced by UA treatment were both inhibited by the ROS inhibitor DPI. Decreased cell viability by UA treatment was also reversed by DPI treatment in the TE-8 and TE-12 cells, suggesting that UA inhibited the growth of ESCC cells by inducing ROS-dependent autophagy, thereby making UA a promising agent for treating ESCC. However, we need to study which ROS subtype is mediated by UA to induce autophagy via Akt-mTOR cell death in ESCC cells. In conclusion, our data indicate that UA induced autophagy by enhancing ROS production via the Akt signaling pathway, a process that ultimately induced cell death in the ESCC cells (Figure 6). These findings suggest that UA is a promising clinical chemotherapeutic agent for treating ESCC.

## 4. Materials and Methods

### 4.1. Reagents and Antibodies

UA was purchased from Cayman Chemicals (Ann Arbor, MI, USA) and dissolved in dimethyl sulfoxide (DMSO) procured from Sigma-Aldrich (St. Louis, MO, USA). Wortmannin and the Akt inhibitors were obtained from Calbiochem (San Diego, CA, USA), and DPI and DCFDA were purchased from Sigma-Aldrich (St. Louis, MO, USA). The LC3 antibody was purchased from Novus Biologicals (Littleton, CO, USA), and antibodies against poly-PARP, caspase-3, cleaved-PARP, cleaved-caspase-3, p62, Akt, p-Akt, and GAPDH were purchased from Cell Signaling Technology (Danvers, MA, USA).

### 4.2. Cell Culture

The human TE-8 and TE-12 esophageal cancer cell lines were obtained from the University of Texas MD Anderson Cancer Center, Houston, TX, USA. These cells were cultured as a monolayer in RPMI1640 medium (Welgene, Gyeongsan, South Korea) supplemented with 10% fetal bovine serum (Gibco Life Technologies, NY, USA) and 1% antibiotic (100 µg/mL penicillin and 100 µg/mL streptomycin) under standard conditions in a humidified atmosphere with 5% carbon dioxide at 37 °C. All experiments were performed with the cells at 60–80% confluence.

### 4.3. Cell Viability Analysis

We determined UA cytotoxicity in the ESCC cells with 3-4,5-dimethylthiazol-2,5-diphenyltetrazolium bromide (MTT) assays, as described previously [66], seeding TE-8 and TE-12 cells onto 96-well plates (Corning, NY, USA). After 24 h of cell seeding, the cells were treated with UA at various concentrations for 48 h. The cells were then incubated with 50 µL MTT solution (2 mg/mL in PBS) for 3 h at 37 °C. After aspirating the media and MTT solution from all wells, 200 µL of DMSO (Sigma-Aldrich, St. Louis, MO, USA) was added to solubilize the formazan crystals for 30 min at room temperature. We measured the absorbance at 570 nm by scanning using an Epoch Microplate Spectrophotometer (BioTek, Winooski, VT, USA). The data are presented as a percentage compared with the untreated control cells.

### 4.4. Soft Agar Colony Formation Assay

We assessed the effects of UA on the ability of ESCC to survive and form colonies in an anchorage-independent medium using a soft agar colony formation assay, as described previously [66,67]. We prepared the bottom layer of soft agar (1%) and prepared the top layer (0.7%) with 5 × 10^4^ cells per well. Cells were exposed to 10, 25, and 50 µM of UA in a 6-well plate and incubated at 37 °C with 5% carbon dioxide for 3 weeks. Colonies were observed under light microscopy, counted, and quantified.

### 4.5. Immunoblot Analysis

We performed immunoblotting, as described previously [67,68]. Briefly, after the cell confluency reached 70%, various UA concentrations were applied to the ESCC cells, which were then harvested after 48 h. Pellets were lysed in ice-cold PRO-PREP^TM^ protein extraction solution (iNtRON Biotechnology, Seoul, South Korea), and the extracts were incubated on ice for 30 min and centrifuged at 13,200 rpm for 20 min at 4 °C. We collected the supernatants and determined the protein concentration using BSA protein assay kits (Pierces Biotechnology, Inc., Rockford, IL, USA). We separated the total protein extract in 10% sodium dodecyl sulfate–polyacrylamide gel electrophoresis and transferred it to a polyvinylidene fluoride membrane (GE Healthcare Life Sciences, Buckinghamshire, UK). We blocked the membrane in 5% milk and incubated it overnight in primary antibodies against PARP, caspase-3, cleaved-PARP, cleaved-caspase-3, LC3, p62, Akt, p-Akt, and glyceraldehyde 3-phosphate dehydrogenase (GAPDH), followed by incubation with horseradish peroxidase-conjugated secondary antibodies. The bands were visualized with the Enhanced Chemiluminescence Kit (Amersham, Arlington Heights, IL, USA).

### 4.6. Wound Healing Assay

The invasion ability assay was performed as described previously [69,70]. Briefly, we performed the wound healing assay with various UA concentrations to determine the effect of UA on cell migration. After cell confluency, we created a scratch using a 200 µL pipette tip through the monolayer culture on a 6-well plate. We then measured the width of the wound over time to evaluate the cancer cells’ migration rate. We repeated the experiments three times.

### 4.7. Matrigel Invasion Assay

BD BioCoatTM MatrigelTM Invasion Chambers (BD Biosciences, San Jose, CA, USA) were used for the in vitro cell invasion assay according to the manufacturer’s protocol. Briefly, the Matrigel-coated chambers were rehydrated in a humidified tissue culture incubator at 37 °C in a 5% CO_2_ atmosphere. Cells (2.5 × 10^4^) were suspended in 500 µL of a medium containing 10% fetal bovine serum in each Matrigel-coated transwell insert, and the lower chamber of the transwell was filled with 500 µL of medium. After incubation, the cultures were washed and stained using the Diff-Quik Kit (Sysmex Corp., Kobe, Japan). Cells on the upper side of the insert membrane were removed, and cells that migrated to the lower side of the membrane were counted on an inverted microscope (magnification, ×100). Five fields were randomly selected, and the invasion rates were calculated as previously described.

### 4.8. Cell Cycle Analysis

The esophageal cancer TE-8 and TE-12 cells were treated with UA (0, 10, 20, 30, 40, or 50 µM) for 48 h in 6-well plate dishes. The cells were harvested and fixed with 75% ethanol at −20 ℃ for 2 h. The cells were stained with propidium iodide (Sigma-Aldrich, St. Louis, MO, USA) at 37 °C for 30 min after fixation. The cell cycle was detected using an FACStar flow cytometer (Becton-Dickinson, San Jose, CA, USA). The sub-G1 phase was analyzed using the BD Accuri™ C6 Software (v1.0.264.21, Accuri Cytometers Inc., Ann Arbor, MI, USA).

### 4.9. FITC Annexin V Staining

After treatment with UA (0, 10, 20, 30, 40, or 50 µM) for 48 h, the TE-8 and TE-12 cells were harvested for the detection of apoptosis cells using the FITC Annexin V Apoptosis Detection Kit II (Becton Dickinson Biosciences, CA, USA). The cells were stained with Annexin V-FITC for 30 min at 37 °C and detected using an FACStar flow cytometer.

### 4.10. GFP Transfection and Fluorescent Microscopy

We transfected the cells (TE-8 and TE-12) with 1 µg of GFP-LC3-expressing plasmid in each well of the 6-well plates using lipofectamine 2000 (Invitrogen, Pittsburgh, PA, USA) according to the manufacturer’s protocol. After 5 h, we treated cells with UA and visualized the GFP or GFP-LC3 fluorescence under a fluorescent microscope [71]. We counted the puncta under a confocal microscope (Nikon, Tokyo, Japan).

### 4.11. Measurement of Intracellular ROS

We loaded the cells (TE-8 and TE-12) with the redox-sensitive dye DCFDA (10 nM) for 20 min at 37 °C and then rinsed the cells twice with Hank’s Balanced Salt Solution and inhibited them with antagonists. We detected DCFDA fluorescence at excitation and emission wavelengths of 488 and 520 nm, respectively. We measured ROS formation using a confocal laser-scanning microscope (Nikon, Tokyo, Japan) and applied DPI treatment (10 µM) and N-acetyl-cysteine (NAC, 2.5 mM) pretreatment for 20 min.

### 4.12. Statistical Analysis

All experiments were performed three times, and the data are presented as mean values with standard error. Comparisons between groups were identified by one-way analysis of variance with Student’s *t*-test. A *p*-value of <0.05 was considered statistically significant.

## Figures and Tables

**Figure 1 ijms-21-09409-f001:**
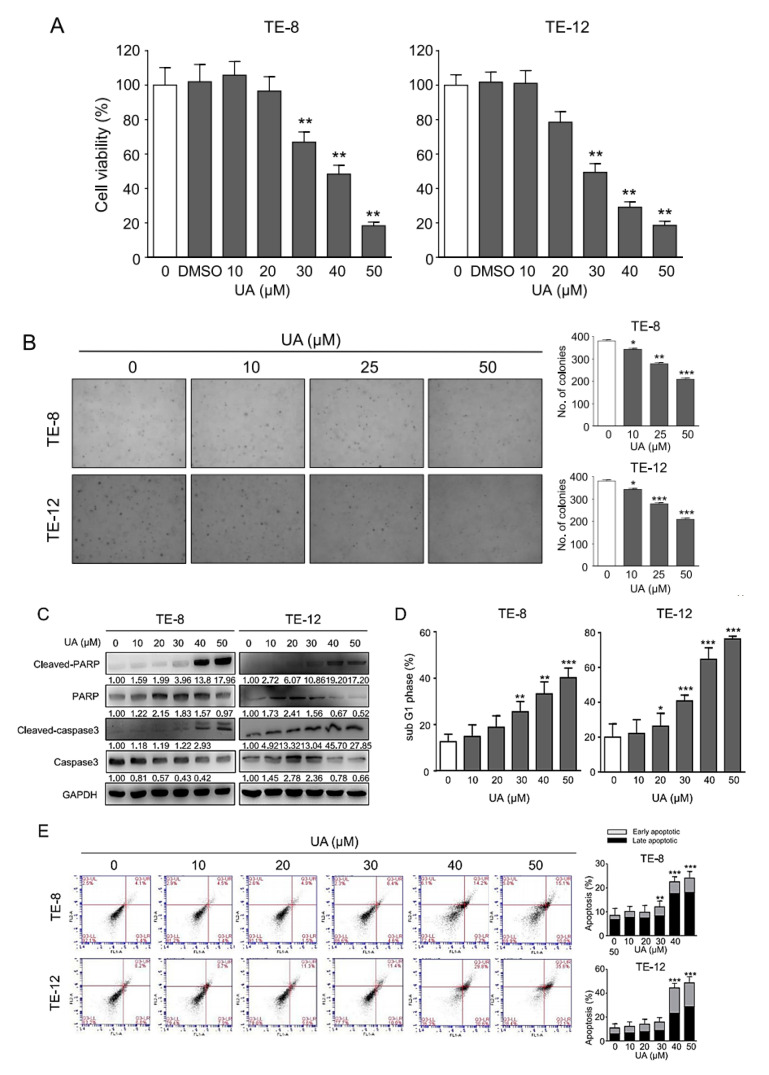
Ursolic acid (UA) inhibits the proliferation of esophageal squamous cell carcinoma (ESCC) cells. (**A**) Effect of UA on cell viability. TE-8 cells and TE-12 cells were treated with UA in a dose-dependent manner. The cells were then assayed with MTT. (**B**) Effect of UA on the colony formation ability of ESCC. The number of colonies formed significantly decreased after UA treatment in a dose-dependent manner in both the TE-8 and TE-12 ESCC cells. Scale bar = 500 µm (**C**) UA induced the expression of cleaved-PARP and cleaved-caspase-3 in the TE-8 cells and TE-12 cells in a dose-dependent manner and decreased the expression of PARP and caspase-3. We employed glyceraldehyde 3-phosphate dehydrogenase as an internal control. (**D**) Induction of sub-G1 phase was analyzed by propidium iodide staining with a flow cytometer. (**E**) Cell apoptosis was detected by fluorescein isothiocyanate–annexin V staining with a flow cytometer. Apoptotic cells were increased in a dose-dependent manner by UA. The data are presented as the mean ± SE for three independent experiments. * *p <* 0.05, ** *p <* 0.01, and *** *p <* 0.001 compared with the control.

**Figure 2 ijms-21-09409-f002:**
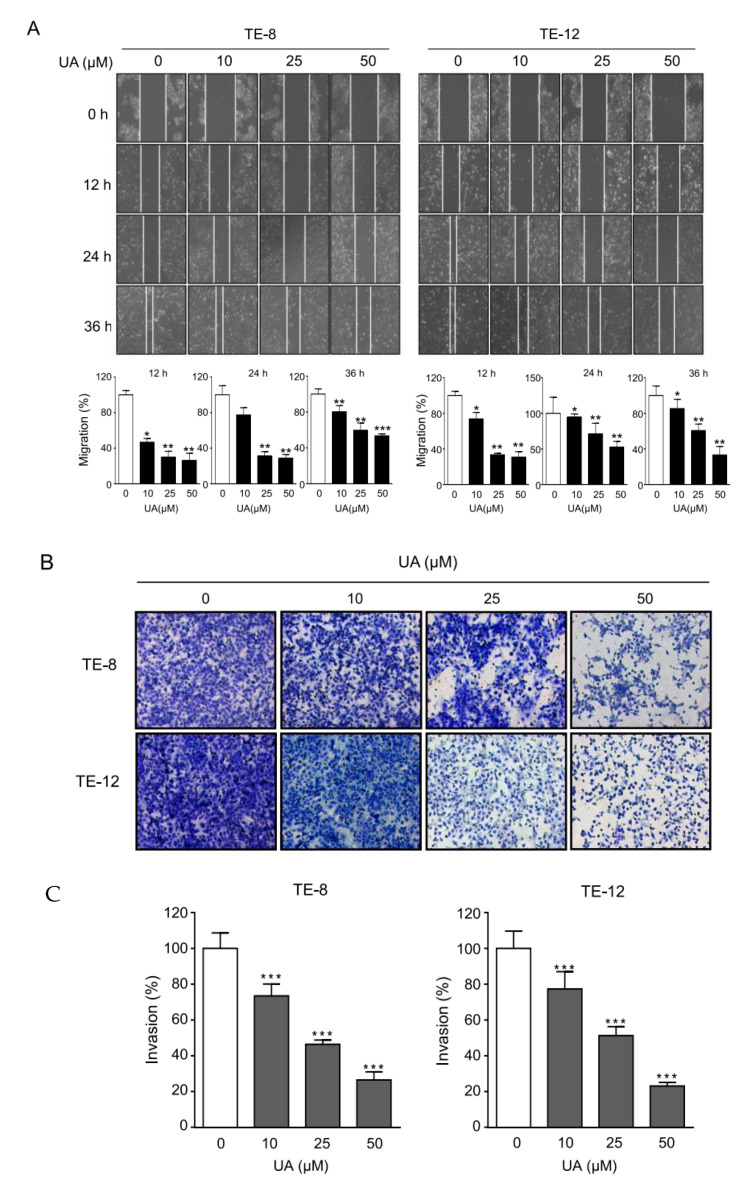
UA inhibits the cell metastasis and invasion of ESCC cells. (**A**) The effect of UA on cell migration by wound healing assay. The migration rate significantly decreased in the cells treated with 25 µM and 50 µM of UA compared with the control. The representative images were obtained at 0, 12, 24, and 36 h. The migration ability was quantified by measuring the gap distance at each time point. (**B**) The effect of UA on the invasive rates in ESCC. TE-8 and TE-12 cells were treated with UA (0, 10, 25, and 50 µM). We detected the invasion rates using a Matrigel invasion assay. (**C**) The data are presented as the mean ± SE for three independent experiments. * *p <* 0.05, ** *p <* 0.01, and *** *p <* 0.001 compared with the control. Scale bar = 100 µm.

**Figure 3 ijms-21-09409-f003:**
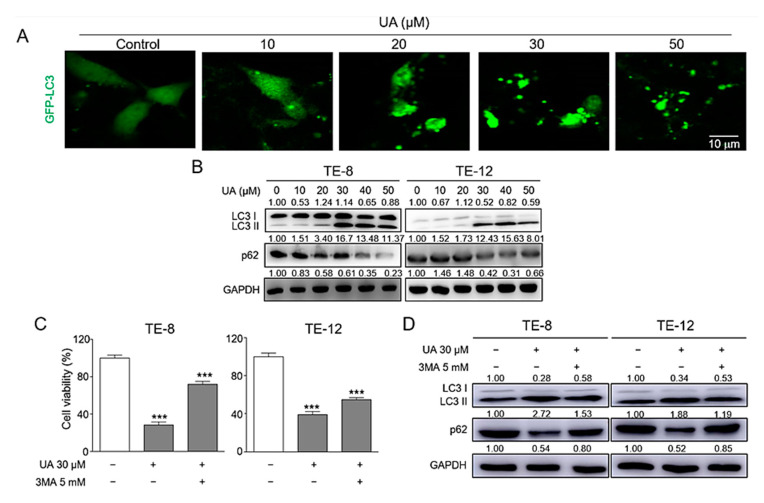
Autophagy was induced by UA in ESCC. (**A**) TE-12 cells transfected by GFP-fused LC3 plasmid and treated with UA. We employed confocal microscopy to visualize the GFP-LC3 puncta present in the cells. (**B**) Western blotting was performed to check the expression of LC3 and p62 after UA treatment. (**C**) Autophagy inhibition by 3-MA caused the induction of cell survival. Pretreatment with the autophagy inhibitor 3-MA (5 mM) for 1 h followed by UA treatment for 48 h resulted in increased ESCC cell survival. (**D**) LC3 and p62 protein expression were detected by Western blot analysis after UA treatment with or without 3-MA. Autophagy inhibition by 3-MA caused LC3 inhibition and p62 protein induction. We employed GAPDH as an internal control. The data are presented as the mean ± SE for three independent experiments. *** *p <* 0.001 compared with the control.

**Figure 4 ijms-21-09409-f004:**
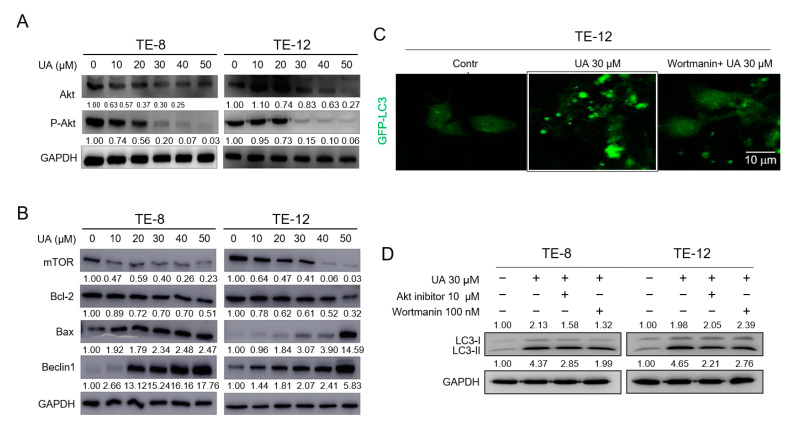
UA induced autophagy in ESCC through the Akt-mTOR signaling pathway. After UA treatment for 48 h, we measured the phosphorylation of Akt (**A**), mTOR, Bcl-2, Bax, and Beclin-1 (**B**) by Western blot analysis. (**C**) We obtained the representative images through confocal microscopy. The GFP-LC3 puncta were more noticeable in the TE-12 cells treated with UA (30 µM) alone compared with the control and the cells treated with a combination of wortmannin and UA (30 µM). (**D**) Pretreatment with wortmannin (100 nM) or Akt inhibitor (10 µM) 2 h prior to UA treatment showed a decrease in LC3 expression. We employed GAPDH as an internal control. The data are presented as the mean ± SE for three independent experiments.

**Figure 5 ijms-21-09409-f005:**
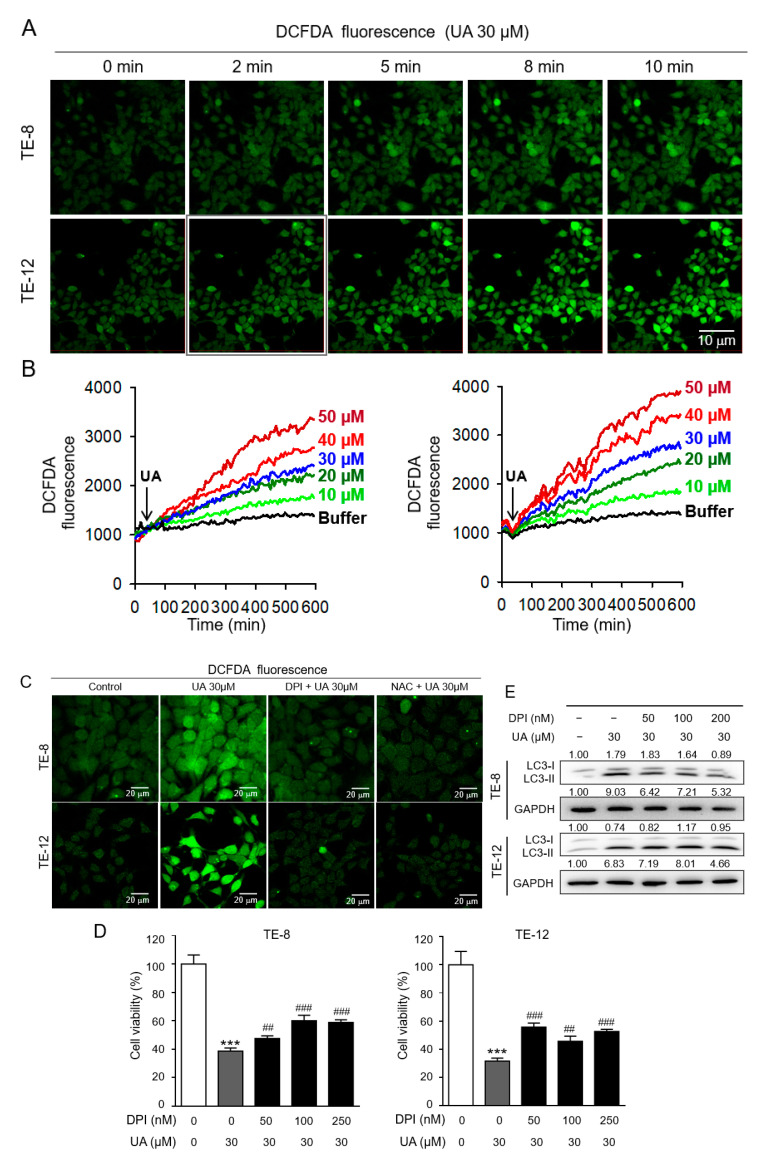
Ursolic acid induced autophagy by reactive oxygen species (ROS) induction in ESCC. (**A**) The UA-induced ROS-mediated DCFDA fluorescence intensity of the cells treated with UA (30 µM) increased in a time-dependent manner. (**B**) The quantitative analysis of ROS by fluorescence-activated cell sorting showed increased DCFDA fluorescence in a dose-dependent manner. (**C**) DPI- and NAC-treated cells showed decreased fluorescence intensity, demonstrating intracellular ROS inhibition. (**D**) Combination treatment with the ROS inhibitor DPI increased cell viability compared with UA treatment alone. (**E**) TE-8 and TE-12 cells were treated with 30 µM UA and 100 nM or 250 nM DPI. We detected LC3 protein levels by Western blot analysis and employed GAPDH as an internal control. The data are presented as the mean ± SE for three independent experiments.*** *p* < 0.001 compared with the control, ^##^
*p* < 0.01, and ^###^
*p* < 0.001 compare with the UA-treated group.

**Figure 6 ijms-21-09409-f006:**
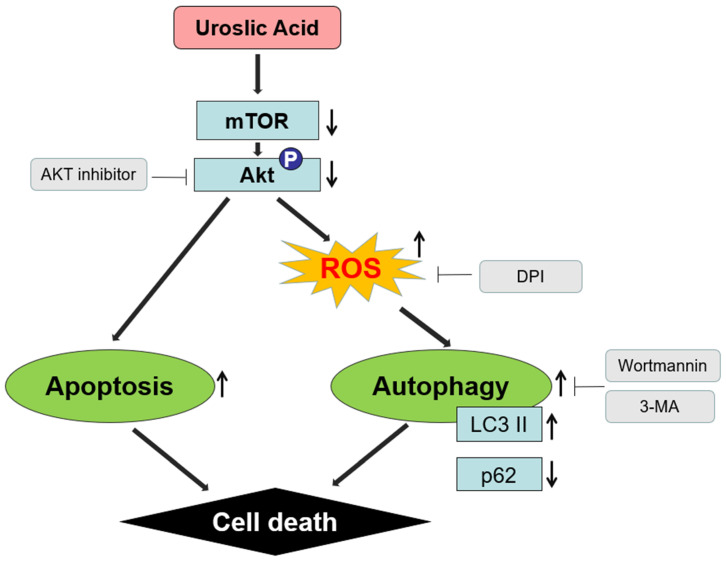
Schematic diagram illustrates the underlying mechanism of UA-induced autophagy through enhanced ROS production via the mTOR/Akt signaling pathway.

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
