# Peer review of "Reactive Oxygen Species-Mediated Autophagy by Ursolic Acid Inhibits Growth and Metastasis of Esophageal Cancer Cells"

_ijms, 2020, doi:10.3390/ijms21249409_

Round 1

Reviewer 1 Report

This study investigated the anticancer mechanisms of ursolic acid in two esophageal squamous cell carcinoma cell lines and revealed that ursolic acid can induce reactive oxygen species-mediated autophagy. This is an interesting finding.

In previous publications, it has been reported that ursolic acid can inhibits cancer cell proliferation, arrest cell cycle progression, and reduces tumorigenesis. It could affect cellular reactive oxygen species, and induce autophagy in cervical cancer and breast cancer cells.

Specific comments:

1 the role of ursolic acid will be more clear and convincing if confirmed in vivo or use primary human cancer cells.

2 Many microscopic images are with low resolution and without scale bars.

3 Compared to diphenyleneiodonium (DPI), N-acetyl-cysteine (NAC) is a more common and straight antioxidant agent.

4 To confirm the role of mTOR/Akt/autophagy pathway, siRNA or gene knock out/down cell line could be used.

Author Response

We appreciate the insightful review of our manuscript by the reviewers of the International Journal of Molecular Science. We have carefully considered each comments and responded accordingly. Point-by-point responses to the reviewers’ comments and revisions made in the manuscript are presented below. Note that all changes in the manuscript are marked in red font (additional content).

We appreciate the insightful review of our manuscript by the reviewers of the International Journal of Molecular Science. We have carefully considered each comments and responded accordingly. Point-by-point responses to the reviewers’ comments and revisions made in the manuscript are presented below. Note that all changes in the manuscript are marked in red font (additional content).

  1. Ref. No.: ijms-990377

Title: Reactive Oxygen Species-Mediated Autophagy by Ursolic Acid Inhibits Esophageal Cancer Cell Growth and Metastasis

Reviewer #1 (Comments to the Author (Required)):

This study investigated the anticancer mechanisms of ursolic acid in two esophageal squamous cell carcinoma cell lines and revealed that ursolic acid can induce reactive oxygen species-mediated autophagy. This is an interesting finding.

In previous publications, it has been reported that ursolic acid can inhibits cancer cell proliferation, arrest cell cycle progression, and reduces tumorigenesis. It could affect cellular reactive oxygen species, and induce autophagy in cervical cancer and breast cancer cells.

Specific comments:

1 the role of ursolic acid will be more clear and convincing if confirmed in vivo or use primary human cancer cells.

Response: Your insightful comments are highly appreciated. We studied the effect of UA (ursolic acid) on gastric tumor growth. As shown, the control group showed a disseminated proliferation of small to medium-sized cells; the cells exhibited varying shapes of hyperchromatic nuclei and scanty cytoplasm. Meanwhile, minimal necrosis and moderate lymphocyte infiltration around the tumor cells were identified in the UA treatment group. These results suggest that UA significantly suppresses gastric xenograft tumor growth in vivo and prevents tumorigenesis. Therefore, we believe that this suppressive effect of UA on xenograft tumor growth in gastric cancer will also be seen in the esophageal cancer cells. Although we did not try to perform in vivo experiments with UA, we will further investigate the effect of UA on esophageal tumor growth.

2 Many microscopic images are with low resolution and without scale bars.

Response: Your insightful comments are highly appreciated. We have revised the images with high quality and labeled the scale bars in all images. For instance, please refer Fig.  1B below.

3 Compared to diphenyleneiodonium (DPI), N-acetyl-cysteine (NAC) is a more common and straight antioxidant agent.

Response: Your insightful comments are highly appreciated. As you suggested, we performed an immuno flurosence experiment after treatment with NAC and UA. As you can see below, NAC’S effects against UA were similar to those of DPI in TE-8 and TE-12 cells. NAC’s antioxidant effects on esophageal cancer cells were similar to those of DPI. WE have added data to Fig. 5C.

4 To confirm the role of mTOR/Akt/autophagy pathway, siRNA or gene knock out/down cell line could be used.

Response: Your insightful comments are highly appreciated. We agree that to confirm the role of the mTOR/Akt/autophagy pathway, siRNA or gene knock out/down cells could be used. Instead of knockdown of Akt, we used two kinds of Akt inhibitors (Akt inhibitor and wortmannin) to inhibit the Mtor/Akt signaling pathway. As shown in Fig. 4C., worthmannin inhibited the LC3 that was induced by UA treatment in TE-12 cells. In addition, according to the western blotting data in Fig. 4D., Akt inhibitor and wortmannin suppressed the LC3II expression that was increased by UA treatment in the TE-8 and TE-12 cells. Therefore, these data support the notion that UA-induced autophagy inhibited cell survival via the Akt-mTOR signaling pathway.

C

Reviewer 2 Report

In this study, Kim and colleagues set out to investigate the molecular mechanism underlying the deleterious effects of ursolic acid against esophageal squamous cell carcinoma (ESCC) cells. The main motivation behind the study is that, although ursolic acid is known to have several anticancer activities (e.g., antiproliferative, proapoptotic, antimetastatic and antiangiogenic), little is known about its molecular mechanism against ESCC (one of the top-10 prevalent cancer types). To do so, authors used several methods, including cell viability and migration assays, colony formation assay, flow cytometry assays, immunoblotting assays, ROS generation assay, and autophagy assays. This set of techniques combined with the use of specific inhibitors is an adequate approach to tackle the objective of the study. The introduction is concise and well-written. Results fit together and are presented in a logical manner. Indeed, the results support the main conclusion of the study, which is that UA increases ROS formation, leading to autophagy and cell death, in ESCC cells. However, key information is missing from the materials and methods and the quality of several figures needs improvement. Please see my concerns numbered below.

  1. Consider calculating the IC50 for each cell type, this would allow other researcher to compare the effects of UA with that of other molecules, as well as other cell types.
  2. In what solvent was the stock UA solution prepared? DMSO? Please provide this information. If the stock solution was made in DMSO, please inform the final concentration of DMSO in the assays.
  3. What is the difference between group “0” and group “DMSO” depicted in figure 1A? Please provide this information.
  4. Please include the time of exposure to UA and the inhibitors at all relevant places. For example, for how long were cells treated in the results shown in figure 3B, C, and D (or figures 4A, B, and D)? Was it 48 h (as in the MTT assay) or 5 h (as in the GFP-LC3 assay)? The lack of this information makes it difficult to understand which happens first, Akt activation or ROS overproduction (see comment 7).
  5. Please describe the procedure used to assess apoptosis and cell cycle by flow cytometry
  6. Please describe the procedure used to perform the Matrigel invasion assay.
  7. Figure 6: The problem with this figure is that Akt can both affect ROS formation and be affected by ROS formation. At lines 84-85 and 263–264, authors indicate that the Akt pathway induces ROS formation, not the other way around, which contradicts the scheme. Since I could not identify the sequence of events shown in the results (see comment 4), I am not sure if the results allow the location of ROS production and Akt signaling in time. Regardless, either the text or the figure needs changes so that they do not contradict each other.
  8. Line 156 and 166 and Figure 3C: It is not clear which method was used to assess cell viability/survival presented in these places. Please provide more details, including the time of exposure.
  9. Line 353: What do you mean with “and stimulated them with an agonist”? Is the “agonist” UA? Please clarify.
  10. With the exception of figures 3 and 4, which have adequate quality, figure quality has room for improvement. There are several issues with the displays:
    1. The resolution is low. I cannot see any cells in figure 1B or 2A. I can hardly read the numbers in figure 1C or 5E, as well as the labels “early apoptotic” and “late apoptotic” in figure 1E. The figure about flow cytometry has very poor resolution, I cannot really see anything.
    2. There is an unwanted gray background in figures 1A and B, 2B, and 5A. There is an unwanted border in figure 1C
  11. There are some languages issues. For example, sentences with unwanted meanings.
    1. Line 22: “UA accumulated vacuoles” means that there were vacuoles in UA, which does not make sense. I think you mean that UA caused accumulation of vacuoles and LC3 puncta in ESCC cells.
    2. Line 25–26: “Cell viability was reversed” also makes no sense. I think you mean that the effect of UA on cell viability was reversed.
    3. Lines 31 and 209: DPI cannot inhibit ROS themselves; it inhibits ROS formation by acting on the enzymes that generate them. I think you mean “a ROS production inhibitor” and “suppressed ROS formation”.

Minor issues

  1. This is a matter of style, but conventionally you should not anticipate results or any implications or discussion in the introduction section. Please consider this.
  2. Line 340: Please cite the reference that “previously described” the assay.
  3. Line 352: There is a problem with the unit in DCFDA concentration. Please check.

Author Response

We appreciate the insightful review of our manuscript by the reviewers of the International Journal of Molecular Science. We have carefully considered each comments and responded accordingly. Point-by-point responses to the reviewers’ comments and revisions made in the manuscript are presented below. Note that all changes in the manuscript are marked in red font (additional content).

Reviewer #2 (Comments to the Author (Required)):

In this study, Kim and colleagues set out to investigate the molecular mechanism underlying the deleterious effects of ursolic acid against esophageal squamous cell carcinoma (ESCC) cells. The main motivation behind the study is that, although ursolic acid is known to have several anticancer activities (e.g., antiproliferative, proapoptotic, antimetastatic and antiangiogenic), little is known about its molecular mechanism against ESCC (one of the top-10 prevalent cancer types). To do so, authors used several methods, including cell viability and migration assays, colony formation assay, flow cytometry assays, immunoblotting assays, ROS generation assay, and autophagy assays. This set of techniques combined with the use of specific inhibitors is an adequate approach to tackle the objective of the study. The introduction is concise and well-written. Results fit together and are presented in a logical manner. Indeed, the results support the main conclusion of the study, which is that UA increases ROS formation, leading to autophagy and cell death, in ESCC cells. However, key information is missing from the materials and methods and the quality of several figures needs improvement. Please see my concerns numbered below.

  1. Consider calculating the IC50 for each cell type, this would allow other researcher to compare the effects of UA with that of other molecules, as well as other cell types.

Response: We appreciate your valuable comment. As you suggested, we used Graph prism 7.0 to calculate the IC50 of the TE-8 and TE-12 esophageal cancer cell line. The IC50 of TE-8 cells was 39.01 μM and that of TE-12 cells was 29.65 μM. We have also added IC50 values of TE-8 and TE-12 cells in the revised manuscript.

  1. In what solvent was the stock UA solution prepared? DMSO? Please provide this information. If the stock solution was made in DMSO, please inform the final concentration of DMSO in the assays.

Response: We appreciate your valuable comment. We dissolved UA into dimethyl sulfoxide (DMSO, #D2660, Sigma-Aldrich, USA). UA stock concentration was 50 μM. When UA concentration was 50 μM, we added 1 μL of UA stock into 2 mL of the medium. Therefore, the DMSO concentration is 0.05% in 50 μM of UA.

  1. What is the difference between group “0” and group “DMSO” depicted in figure 1A? Please provide this information.

Response: We appreciate your valuable comment. Group “0” means only medium, group “DMSO” means DMSO is added to the medium.

  1. Please include the time of exposure to UA and the inhibitors at all relevant places. For example, for how long were cells treated in the results shown in figure 3B, C, and D (or figures 4A, B, and D)? Was it 48 h (as in the MTT assay) or 5 h (as in the GFP-LC3 assay)? The lack of this information makes it difficult to understand which happens first, Akt activation or ROS overproduction (see comment 7).

Response: We appreciate your valuable comment. In Fig. 3 and Fig. 4, drug treatment time is described as follows: 3-MA (5 mM) pretreatment was given for 1 h before UA (30 μM) treatment for 48 h. Wortmannin (100 nM) pretreatment was given for 2 h before UA (30 μM) treatment for 48 h.

  1. Please describe the procedure used to assess apoptosis and cell cycle by flow cytometry

Response: We appreciate your valuable comment. We described cell cycle analysis in the manuscript. For the cell cycle, the TE-8 and TE-12 esophageal cancer cells were treated with ursolic acid (0, 10, 20, 30, 40, or 50 µM) for 48 h in a 6-well plate dish. The cells were then harvested and fixed with 75% ethanol at −20°C for 2 h. The cells were then stained with propidium iodide (Sigma Chemicals, St. Louis, MO, USA) at 37°C 30 min after fixation. The cell cycle was detected using an FACStar flow cytometer (Becton-Dickinson, San Jose, CA, USA).The subG1 phase was analyzed using the BD Accuri™ C6 Software (Version 1.0.264.21, Accuri Cytometers Inc., Ann Arbor, Michigan, United States).

For FITC Annexin V staining, after treatment with UA (0, 10, 20, 30, 40, or 50 µM) for 48 h, the TE-8 and TE-12 cells were harvested to determine apoptotic cells using the FITC Annexin V Apoptosis Detection Kit II (Becton Dickinson Biosciences, CA, USA). The cells were stained with Annexin V-FITC for 30 min at 37°C and detected using an FACStar flow cytometer (Becton-Dickinson, San Jose, CA, USA).

As you suggested, we have added these details to the revised manuscript.

  1. Please describe the procedure used to perform the Matrigel invasion assay.

Response: We appreciate your valuable comment. For the Matrigel invasion assay, BD BioCoatTM MatrigelTM Invasion Chambers (BD Biosciences, San Jose, CA, USA) were used for the in vitro cell invasion assay according to the manufacturer’s protocol. Briefly, the Matrigel-coated chambers were rehydrated in a humidified tissue culture incubator at 37°C in a 5% CO2 atmosphere. The cells (2.5×104) were suspended in 500 μL of the medium containing 10% fetal bovine serum in each Matrigel-coated transwell insert, and the lower chamber of the transwell was filled with 500 μl of the medium. After incubation, the cultures were washed and stained with the Diff-Quik Kit (Sysmex Corp., Kobe, Japan). The cells on the upper side of the insert membrane were removed, and those that migrated to the lower side of the membrane were counted under an inverted microscope (magnification, ×100). Five fields were randomly selected, and the invasion rates were calculated as previously described. As you suggested, we have added this information to the revised manuscript.

  1. Figure 6: The problem with this figure is that Akt can both affect ROS formation and be affected by ROS formation. At lines 84-85 and 263–264, authors indicate that the Akt pathway induces ROS formation, not the other way around, which contradicts the scheme. Since I could not identify the sequence of events shown in the results (see comment 4), I am not sure if the results allow the location of ROS production and Akt signaling in time. Regardless, either the text or the figure needs changes so that they do not contradict each other.

Response: We appreciate your valuable comment. As you mentioned, there were contradictions between the scheme of Fig.6 and the description in the manuscript. We have revised the scheme in Fig. 6 for consistency with the description in the manuscript.

  1. Line 156 and 166 and Figure 3C: It is not clear which method was used to assess cell viability/survival presented in these places. Please provide more details, including the time of exposure.

Response: We appreciate your valuable comment. We performed the MTT assay to detect cell viability and cell survival. Pre-treatment with autophagy inhibitor 3-MA (5 mM) was performed for 1 h followed by a 48-h UA treatment.

  1. Line 353: What do you mean with “and stimulated them with an agonist”? Is the “agonist” UA? Please clarify.

Response: We appreciate your valuable comment. As you suggested, we used the ROS antagonist. The sentence was poorly phrased. We used two kinds of ROS antagonists, DPI and NAC, for inhibition of UA-induced ROS in esophageal cancer cells. We have revised this sentence in the revised manuscript.

  1. With the exception of figures 3 and 4, which have adequate quality, figure quality has room for improvement. There are several issues with the displays:

  1. The resolution is low. I cannot see any cells in figure 1B or 2A. I can hardly read the numbers in figure 1C or 5E, as well as the labels “early apoptotic” and “late apoptotic” in figure 1E. The figure about flow cytometry has very poor resolution, I cannot really see anything.

Response: We appreciate your valuable comment. As you suggested, we have improved the labeling and figure quality and inserted high quality figures in the revised manuscript.

  1. There is an unwanted gray background in figures 1A and B, 2B, and 5A. There is an unwanted border in figure 1C

Response: We appreciate your valuable comment. As you suggested, we have improved the figure quality and inserted high quality figures in the manuscript.

  1. There are some languages issues. For example, sentences with unwanted meanings.

Response: We appreciate your valuable comment. We will request English language editing services to address the languages issues.

  1. Line 22: “UA accumulated vacuoles” means that there were vacuoles in UA, which does not make sense. I think you mean that UA caused accumulation of vacuoles and LC3 puncta in ESCC cells.

Response: We appreciate your valuable comment. As you recommended, we have revised the sentence in the revised manuscript.

  1. Line 25–26: “Cell viability was reversed” also makes no sense. I think you mean that the effect of UA on cell viability was reversed.

Response: We appreciate your valuable comment. As you suggested, we have revised the sentence in the revised manuscript.

  1. Lines 31 and 209: DPI cannot inhibit ROS themselves; it inhibits ROS formation by acting on the enzymes that generate them. I think you mean “a ROS production inhibitor” and “suppressed ROS formation”.

Response: We appreciate your valuable comment. As you suggested, we have addressed these language issues in the revised manuscript.

Minor issues

  1. This is a matter of style, but conventionally you should not anticipate results or any implications or discussion in the introduction section. Please consider this. 
  2. Response: We appreciate your valuable comment. We will take your recommendation into consideration.
  3. Line 340: Please cite the reference that “previously described” the assay. 
  4. Response: We appreciate your valuable comment. As you suggested, we have added the reference to the revised manuscript.
  5. Line 352: There is a problem with the unit in DCFDA concentration. Please check.
  6. Response: We appreciate your valuable comment. We have changed the unit of DCFDA concentration in the revised manuscript.

Round 2

Reviewer 1 Report

Need to highlight the revisions regarding to the languages issues in the manuscript.